# REVISITING SELF-TRAINING
# FOR NEURAL SEQUENCE GENERATION

**Junxian He**[*]
Carnegie Mellon University
junxianh@cs.cmu.edu

**Jiatao Gu**[*]**, Jiajun Shen, Marc'Aurelio Ranzato**
Facebook AI Research, New York, NY
{jgu,jiajunshen,ranzato}@fb.com

## ABSTRACT

Self-training is one of the earliest and simplest semi-supervised methods. The key idea is to augment the original labeled dataset with unlabeled data paired with the model's prediction (i.e. the *pseudo-parallel* data). While self-training has been extensively studied on classification problems, in complex sequence generation tasks (e.g. machine translation) it is still unclear how self-training works due to the compositionality of the target space. In this work, we first empirically show that self-training is able to decently improve the supervised baseline on neural sequence generation tasks. Through careful examination of the performance gains, we find that the perturbation on the hidden states (i.e. dropout) is critical for self-training to benefit from the pseudo-parallel data, which acts as a regularizer and forces the model to yield close predictions for similar unlabeled inputs. Such effect helps the model correct some incorrect predictions on unlabeled data. To further encourage this mechanism, we propose to inject noise to the input space, resulting in a "noisy" version of self-training. Empirical study on standard machine translation and text summarization benchmarks shows that noisy self-training is able to effectively utilize unlabeled data and improve the performance of the supervised baseline by a large margin.[1]

## 1 INTRODUCTION

Deep neural networks often require large amounts of labeled data to achieve good performance. However, acquiring labels is a costly process, which motivates research on methods that can effectively utilize unlabeled data to improve performance. Towards this goal, semi-supervised learning (Chapelle et al., 2009) methods that take advantage of both labeled and unlabeled data are a natural starting point. In the context of sequence generation problems, semi-supervised approaches have been shown to work well in some cases. For example, back-translation (Sennrich et al., 2015) makes use of the monolingual data on the target side to improve machine translation systems, latent variable models (Kingma et al., 2014) are employed to incorporate unlabeled source data to facilitate sentence compression (Miao & Blunsom, 2016) or code generation (Yin et al., 2018).

In this work, we revisit a much older and simpler semi-supervised method, self-training (ST, Scudder (1965)), where a base model trained with labeled data acts as a "teacher" to label the unannotated data, which is then used to augment the original small training set. Then, a "student" model is trained with this new training set to yield the final model. Originally designed for classification problems, common wisdom suggests that this method may be effective only when a good fraction of the predictions on unlabeled samples are correct, otherwise mistakes are going to be reinforced (Zhu & Goldberg, 2009). In the field of natural language processing, some early work have successfully applied self-training to word sense disambiguation (Yarowsky, 1995) and parsing (McClosky et al., 2006; Reichart & Rappoport, 2007; Huang & Harper, 2009).

However, self-training has not been studied extensively when the target output is natural language. This is partially because in language generation applications (e.g. machine translation) hypotheses are often very far from the ground-truth target, especially in low-resource settings. It is natural to

---

[*]Equal Contribution. Most of the work is done during Junxian's internship at FAIR.
[1]Code is available at https://github.com/jxhe/self-training-text-generation.

---

**Algorithm 1** Classic Self-training

---

1: Train a base model $f_{\boldsymbol{\theta}}$ on $L = \{\boldsymbol{x}_i, \boldsymbol{y}_i\}_{i=1}^l$
2: **repeat**
3:     Apply $f_{\boldsymbol{\theta}}$ to the unlabeled instances $U$
4:     Select a subset $S \subset \{(\boldsymbol{x}, f_{\boldsymbol{\theta}}(\boldsymbol{x})) | \boldsymbol{x} \in U\}$
5:     Train a new model $f_{\boldsymbol{\theta}}$ on $S \cup L$
6: **until** convergence or maximum iterations are reached

---

ask whether self-training can be useful at all in this case. While Ueffing (2006) and Zhang & Zong (2016) explored self-training in statistical and neural machine translation, only relatively limited gains were reported and, to the best of our knowledge, it is still unclear what makes self-training work. Moreover, Zhang & Zong (2016) did not update the decoder parameters when using pseudo parallel data noting that "*synthetic target parts may negatively influence the decoder model of NMT*".

In this paper, we aim to answer two questions: (1) How does self-training perform in sequence generation tasks like machine translation and text summarization? Are "bad" pseudo targets indeed catastrophic for self-training? (2) If self-training helps improving the baseline, what contributes to its success? What are the important ingredients to make it work?

Towards this end, we first evaluate self-training on a small-scale machine translation task and empirically observe significant performance gains over the supervised baseline (§3.2), then we perform a comprehensive ablation analysis to understand the key factors that contribute to its success (§3.3). We find that the decoding method to generate pseudo targets accounts for part of the improvement, but more importantly, the perturbation of hidden states – dropout (Hinton et al., 2012) – turns out to be a crucial ingredient to prevent self-training from falling into the same local optimum as the base model, and this is responsible for most of the gains. To understand the role of such noise in self-training, we use a toy experiment to analyze how noise effectively propagates labels to nearby inputs, sometimes helping correct incorrect predictions (§4.1). Motivated by this analysis, we propose to inject additional noise by perturbing also the input. Comprehensive experiments on machine translation and text summarization tasks demonstrate the effectiveness of noisy self-training.

## 2 SELF-TRAINING

Formally, in conditional sequence generation tasks like machine translation, we have a parallel dataset $L = \{\boldsymbol{x}_i, \boldsymbol{y}_i\}_{i=1}^l$ and a large unlabeled dataset $U = \{\boldsymbol{x}_j\}_{j=l+1}^{l+u}$, where $|U| > |L|$ in most cases. As shown in Algorithm 1, classic self-training starts from a base model trained with parallel data $L$, and iteratively applies the current model to obtain predictions on unlabeled instances $U$, then it incorporates a subset of the *pseudo parallel* data $S$ to update the current model.

There are two key factors: (1) Selection of the subset $S$. $S$ is usually selected based on some confidence scores (e.g. log probability) (Yarowsky, 1995) but it is also possible for $S$ to be the whole pseudo parallel data (Zhu & Goldberg, 2009). (2) Combination of real and pseudo parallel data. A new model is often trained on the two datasets jointly as in back-translation, but this introduces an additional hyper-parameter to weigh the importance of the parallel data relative to the pseudo data (Edunov et al., 2018). Another way is to treat them separately – first we train the model on pseudo parallel data $S$, and then fine-tune it on real data $L$. In our preliminary experiments, we find that the *separate* training strategy with the *whole* pseudo parallel dataset (i.e. $S = \{(\boldsymbol{x}, f_{\boldsymbol{\theta}}(\boldsymbol{x})) | \boldsymbol{x} \in U\}$) produces better or equal performance for neural sequence generation while being simpler. Therefore, in the remainder of this paper we use this simpler setting. We include quantitative comparison regarding joint training, separate training, and pseudo-parallel data filtering in Appendix B, where separate training is able to match (or surpass) the performance of joint training.

In self-training, the unsupervised loss $\mathcal{L}_U$ from unlabeled instances is defined as:

$$\mathcal{L}_U = -\mathbb{E}_{\boldsymbol{x} \sim p(\boldsymbol{x})} \mathbb{E}_{\boldsymbol{y} \sim p_{\boldsymbol{\theta}^*}(\boldsymbol{y}|\boldsymbol{x})} \log p_{\boldsymbol{\theta}}(\boldsymbol{y}|\boldsymbol{x}), \tag{1}$$

where $p(\boldsymbol{x})$ is the empirical data distribution approximated with samples from $S$, $p_{\boldsymbol{\theta}}(\boldsymbol{y}|\boldsymbol{x})$ is the conditional distribution defined by the model. $\boldsymbol{\theta}^*$ is the parameter from the last iteration (initially it

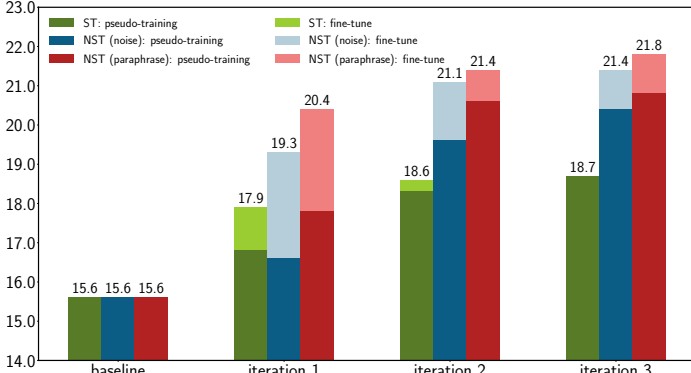

Figure 1: BLEU on WMT100K dataset from the supervised baseline and different self-training variants. We plot the results over 3 iterations. "ST" denotes self-training while "NST" denotes noisy self training.

| Methods | PT | FT |
|---|---|---|
| baseline | – | 15.6 |
| ST (scratch) | 16.8 | 17.9 |
| ST (baseline) | 16.5 | 17.5 |

Table 1: Test tokenized BLEU on WMT100K. Self-training results are from the first iteration. "Scratch" denotes that the system is initialized randomly and trained from scratch, while "baseline" means it is initialized with the baseline model.

is set as the parameter of the supervised baseline), and fixed within the current iteration. Eq. 1 reveals the connection between self-training and entropy regularization (Grandvalet & Bengio, 2005). In the context of classification, self-training can be understood from the view of entropy regularization (Lee, 2013), which favors a low-density separation between classes, a commonly assumed prior for semi-supervised learning (Chapelle & Zien, 2005).

# 3 A CASE STUDY ON MACHINE TRANSLATION

To examine the effectiveness of self-training on neural sequence generation, we start by analyzing a machine translation task. We then perform ablation analysis to understand the contributing factors of the performance gains.

## 3.1 SETUP

We work with the standard WMT 2014 English-German dataset consisting of about 3.9 million training sentence pairs after filtering long and imbalanced pairs. Sentences are encoded using 40K byte-pair codes (Sennrich et al., 2016). As a preliminary experiment, we randomly sample 100K sentences from the training set to train the model and use the remaining English sentences as the unlabeled monolingual data. For convenience, we refer to this dataset as WMT100K. Such synthetic setting allows us to have high-quality unlabeled data to verify the performance of self-training. We train with the Base Transformer architecture (Vaswani et al., 2017) and dropout rate at 0.3. Full training and optimization parameters can be found in Appendix A.1. All experiments throughout this paper including the transformer implementation are based on the fairseq toolkit (Ott et al., 2019), and all results are in terms of case-sensitive tokenized BLEU (Papineni et al., 2002). We use beam search decoding (beam size 5) to create the pseudo targets and to report BLEU on test set.

## 3.2 OBSERVATIONS

In Figure 1, we use green bars to show the result of applying self-training for three iterations. We include both (1) *pseudo-training (PT)*: the first step of self-training where we train a new model (from scratch) using *only* the pseudo parallel data generated by the current model, and (2) *fine-tuning (FT)*: the fine-tuned system using real parallel data based on the pretrained model from the PT step. Note that in the fine-tuning step the system is re-initialized from scratch. Surprisingly, we find that the pseudo-training step at the first iteration is able to improve BLEU even if the model is only trained on its own predictions, and fine-tuning further boosts the performance. The test BLEU keeps improving over the first three iterations, until convergence to outperform the initial baseline by 3 BLEU points.

| Methods | PT | FT |
|---|---|---|
| baseline | – | 15.6 |
| baseline (w/o dropout) | – | 5.2 |
| ST (beam search, w/ dropout) | 16.5 | 17.5 |
| ST (sampling, w/ dropout) | 16.1 | 17.0 |
| ST (beam search, w/o dropout) | 15.8 | 16.3 |
| ST (sampling, w/o dropout) | 15.5 | 16.0 |
| Noisy ST (beam search, w/o dropout) | 15.8 | 17.9 |
| Noisy ST (beam search, w/ dropout) | **16.6** | **19.3** |

Table 2: Ablation study on WMT100K data. For ST and noisy ST, we initialize the model with the baseline and results are from one single iteration. Dropout is varied only in the PT step, while dropout is always applied in FT step. Different decoding methods refer to the strategy used to create the pseudo target. At test time we use beam search decoding for all models.

This behaviour is unexpected because no new information seems to be injected during this iterative process – target sentences of the monolingual data are from the base model's predictions, thus translation errors are likely to remain, if not magnified. This is different from back-translation where new knowledge may originate from an additional backward translation model and real monolingual targets may help the decoder generate more fluent sentences.

One straightforward hypothesis is that the added pseudo-parallel data might implicitly change the training trajectory towards a (somehow) better local optimum, given that we train a new model *from scratch* at each iteration. To rule out this hypothesis, we perform an ablation experiment and initialize $\boldsymbol{\theta}$ from the last iteration (i.e. $\boldsymbol{\theta}^*$). Formally, based on Eq. 1 we have:

$$\nabla_{\boldsymbol{\theta}} \mathcal{L}_U |_{\boldsymbol{\theta}=\boldsymbol{\theta}^*} = -\mathbb{E}_{\boldsymbol{x} \sim p(\boldsymbol{x})} \big[ \nabla_{\boldsymbol{\theta}} \mathbb{E}_{\boldsymbol{y} \sim p_{\boldsymbol{\theta}^*}(\boldsymbol{y}|\boldsymbol{x})} \log p_{\boldsymbol{\theta}}(\boldsymbol{y}|\boldsymbol{x}) |_{\boldsymbol{\theta}=\boldsymbol{\theta}^*} \big] = 0, \tag{2}$$

because the conditional log likelihood is maximized when $p_{\boldsymbol{\theta}}(\boldsymbol{y}|\boldsymbol{x})$ matches the underlying data distribution $p_{\boldsymbol{\theta}^*}(\boldsymbol{y}|\boldsymbol{x})$. Therefore, the parameter $\boldsymbol{\theta}$ should not (at least not significantly) change if we initialize it with $\boldsymbol{\theta}^*$ from the last iteration.

Table 1 shows the comparison results of these two initialization schemes at the first iteration. Surprisingly, continuing training from the baseline model also yields an improvement of 1.9 BLEU points, comparable to initializing from random. While stochastic optimization introduces randomness in the training process, it is startling that continuing training gives such a non-trivial improvement. Next, we investigate the underlying reasons for this.

### 3.3 The Secret Behind Self-Training

To understand why continuing training contradicts Eq. 2 and improves translation performance, we examine possible discrepancies between our assumptions and the actual implementation, and formulate two new hypotheses:

**H1. Decoding Strategy.** According to this hypothesis, the gains come from the use of beam search for decoding unlabeled data. Since our focus is a sequence generation task, we decode $\boldsymbol{y}$ with beam search to approximate the expectation in $\mathbb{E}_{\boldsymbol{y} \sim p_{\boldsymbol{\theta}^*}(\boldsymbol{y}|\boldsymbol{x})} \log p_{\boldsymbol{\theta}}(\boldsymbol{y}|\boldsymbol{x})$, yielding a biased estimate, while sampling decoding would result in an unbiased Monte Carlo estimator. The results in Table 2 demonstrate that the performance drops by 0.5 BLEU when we change the decoding strategy to sampling, which implies that beam search does contribute a bit to the performance gains. This phenomenon makes sense intuitively since beam search tends to generate higher-quality pseudo targets than sampling, and the subsequent cross-entropy training might benefit from implicitly learning the decoding process. However, the decoding strategy hypothesis does not fully explain it, as we still observe a gain of 1.4 BLEU points over the baseline from sampling decoding with dropout.

**H2. Dropout (Hinton et al., 2012).** Eq. 1 and Eq. 2 implicitly ignore a (seemingly) small difference between the model used to produce the pseudo targets and the model used for training: at test/decoding time the model does not use dropout while at training time dropout noise is injected in the model hidden states. At training time, the model is forced to produce the same (pseudo) targets given the same set of inputs and the same parameter set but various noisy versions of the

hidden states. The conjecture is that the additional expectation over dropout noise renders Eq. 2 false. To verify this, we remove dropout in the pseudo training step[2]. The results in Table 2 indicate that without dropout the performance of beam search decoding drops by 1.2 BLEU, just 0.7 BLEU higher than the baseline. Moreover, the pseudo-training performance of sampling without dropout is almost the same as the baseline, which finally agrees with our intuitions from Eq. 2.

In summary, Table 2 suggests that beam-search decoding contributes only partially to the performance gains, while the implicit perturbation – dropout – accounts for most of it. However, it is still mysterious why such perturbation results in such large performance gains. If dropout is meant to avoid overfitting and fit the target distribution better in the pseudo-training step, why does it bring advantages over the baseline given that the target distribution is from the baseline model itself ? This is the subject of the investigation in the next section.

# 4 NOISE IN SELF-TRAINING

## 4.1 THE ROLE OF NOISE

One hypothesis as to why noise (perturbation) is beneficial for self-training, is that it enforces local smoothness for this task, that is, semantically similar inputs are mapped to the same or similar targets. Since the assumption that similar input should ideally produce similar target largely holds for most tasks in practice, this smoothing effect of pseudo-training step may provide a favorable regularization for the subsequent fine-tuning step. Unlike standard regularization in supervised training which is local to the real parallel data, self-training smooths the data space covered by the additional and much larger monolingual data.

| Methods | smoothness | symmetric | error |
|---------|-----------|-----------|-------|
| baseline | 9.1 | 9.8 | 7.6 |
| ST | 8.2 | 9.0 | 6.2 |
| noisy ST | **7.3** | **8.2** | **4.5** |

Table 3: Results on the toy sum dataset. For ST and noisy ST, smoothness ($\downarrow$) and symmetric ($\downarrow$) results are from the pseudo-training step, while test errors ($\downarrow$) are from fine-tuning, all at the first iteration.

To verify this hypothesis more easily, we work with the toy task of summing two integers in the range 0 to 99. We concatenate the two integers and view them as a sequence of digits, the sum is also predicted at the digit level, thus this is still a sequence to sequence task. There are 10000 possible data points in the entire space, and we randomly sample 250 instances for training,[3] 100 for validation, 5000 for test, and 4000 as the unlabeled data. Test errors are computed as the absolute difference between the predicted integer and the ground-truth integer. We use an LSTM model to tackle this task. We perform self-training for one iteration on this toy sum dataset and initialize the model with the base model to rule out differences due to the initialization. Setup details are in Appendix A.1.

For any integer pair $(x_1, x_2)$, we measure local smoothness as the standard deviation of the predictions in a $3 \times 3$ neighborhood of $(x_1, x_2)$. These values are averaged over all the 10000 points to obtain the overall smoothness. We compare smoothness between baseline and ST pseudo-training in Table 3. To demonstrate the effect of smoothing on the fine-tuning step, we also report test errors after fine-tuning. We observe that ST pseudo-training attains better smoothness, which helps reducing test errors in the subsequent fine-tuning step.

One natural question is whether we could further improve performance by encouraging even lower smoothness value, although there is a clear trade-off, as a totally smooth model that outputs a constant value is also a bad predictor. One way to decrease smoothness is by increasing the dropout probability in the pseudo-training step, but a large dropout (like 0.5) makes the model too unstable and slow at converging. Therefore, we consider a simple model-agnostic perturbation process – perturbing the input, which we refer to as *noisy self-training* (noisy ST).

---

[2] During finetuning, we still use dropout.

[3] We choose 250 instances since we find that 500 training samples already yields perfect performance on this task. However, we want to mimic real seq2seq tasks where the supervised models are often far from perfect.

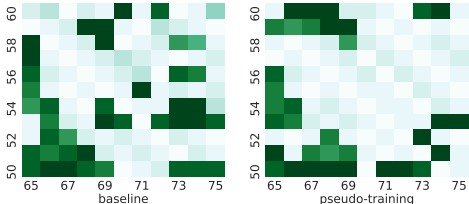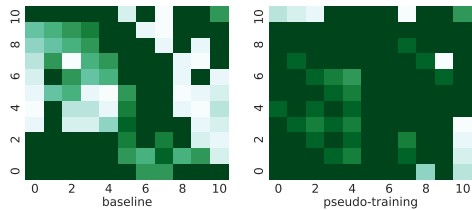

Figure 2: Two examples of error heat map on the toy sum dataset that shows the effect of smoothness. The left panel of each composition is from the baseline, and the right one is from the pseudo-training step at the first iteration. $x$ and $y$ axes represent the two input integers. Deeper color represent larger errors.

## 4.2 NOISY SELF-TRAINING

If we perturb the input during the pseudo-training step, then Eq. 1 would be modified to:

$$\mathcal{L}_U = -\mathbb{E}_{\boldsymbol{x'} \sim g(\boldsymbol{x}), \boldsymbol{x} \sim p(\boldsymbol{x})} \mathbb{E}_{\boldsymbol{y} \sim p_{\boldsymbol{\theta}*}(\boldsymbol{y}|\boldsymbol{x})} \log p_{\boldsymbol{\theta}}(\boldsymbol{y}|\boldsymbol{x'}), \tag{3}$$

where $g(\boldsymbol{x})$ is a perturbation function. Note that we apply both input perturbation and dropout in the pseudo-training step for noisy ST throughout the paper, but include ablation analysis in §4.3. We first validate noisy ST in the toy sum task. We shuffle the two integers in the input as the perturbation function. Such perturbation is suitable for this task since it would help the model learn the commutative law as well. To check that, we also measure the symmetry of the output space. Specifically, for any point $(x_1, x_2)$, we compute $|f(x_1, x_2) - f(x_2, x_1)|$ and average it over all the points. Both smoothness and symmetry values are reported in Table 3. While we do not explicitly perturb the input at nearby integers, the shuffling perturbation greatly improves the smoothness metric as well. Furthermore, predictions are more symmetric and test errors are reduced.

In order to illustrate the effect of smoothness, in Figure 2 we show two examples of error heat map.[4] When a point with large error is surrounded by points with small errors, the labels might propagate due to smoothing and its error is likely to become smaller, resulting in a "self-correcting" behaviour, as demonstrated in the left example of Figure 2. However, the prediction of some points might become worse due to the opposite phenomenon too, as shown in the right example of Figure 2. Therefore, the smoothing effect by itself does not guarantee a performance gain in the pseudo-training step, but fine-tuning benefits from it and seems to consistently improve the baseline in all datasets we experiment with.

## 4.3 OBSERVATIONS ON MACHINE TRANSLATION

Next, we apply noisy self-training to the more realistic WMT100 translation task. We try two different perturbation functions: (1) Synthetic noise as used in unsupervised MT (Lample et al., 2018), where the input tokens are randomly dropped, masked, and shuffled. We use the default noising parameters as in unsupervised MT but study the influence of noise level in §5.4. (2) Paraphrase. We translate the source English sentences to German and translate it back to obtain a paraphrase as the perturbation. Figure 1 shows the results over three iterations. Noisy ST (NST) greatly outperforms the supervised baseline by over 6 BLEU points and normal ST by 3 BLEU points, while synthetic noise does not exhibit much difference from paraphrasing. Since synthetic noise is much simpler and more general, in the remaining experiments we use synthetic noise unless otherwise specified.

Next, we report an ablation analysis of noisy ST when removing dropout at the pseudo-training step in Table 2. Noisy ST without dropout improves the baseline by 2.3 BLEU points and is comparable to normal ST with dropout. When combined together, noisy ST with dropout produces another 1.4 BLEU improvement, indicating that the two perturbations are complementary.

---

[4]Error heat map for the entire space can be found in Appendix C.

| Methods | WMT English-German | | FloRes English-Nepali | | |
| --- | --- | --- | --- | --- | --- |
| | 100K (+3.8M mono) | 3.9M (+20M mono) | En-Origin | Ne-Origin | Overall |
| baseline | 15.6 | 28.3 | 6.7 | 2.3 | 4.8 |
| BT | 20.5 | – | 8.2 | **4.5** | **6.5** |
| noisy ST | **21.4** | **29.3** | **8.9** | 3.5 | **6.5** |

Table 4: Results on two machine translation datasets. For WMT100K, we use the remaining 3.8M English and German sentences from training data as unlabeled data for noisy ST and BT, respectively.

## 5 EXPERIMENTS

Our experiments below are designed to examine whether the noisy self-training is generally useful across different sequence generation tasks and resource settings. To this end, we conduct experiments on two machine translation datasets and one text summarization dataset to test the effectiveness under both high-resource and low-resource settings.

### 5.1 GENERAL SETUP

We run noisy self-training for three iterations or until performance converges. The model is trained from scratch in the pseudo-training step at each iteration since we found this strategy to work slightly better empirically. Full model and training details for all the experiments can be found in Appendix A.1. In some settings, we also include back-translation (BT, Sennrich et al., 2015) as a reference point, since this is probably the most successful semi-supervised learning method for machine translation. However, we want to emphasize that BT is not directly comparable to ST since they use different resources (ST utilizes the unlabeled data on the *source* side while BT leverages *target* monolingual data) and use cases. For example, BT is not very effective when we translate English to extremely low-resource languages where there is almost no in-domain target monolingual data available. We follow the practice in (Edunov et al., 2018) to implement BT where we use unrestricted sampling to translate the target data back to the source. Then, we train the real and pseudo parallel data jointly and tune the upsampling ratio of real parallel data.

### 5.2 MACHINE TRANSLATION

We test the proposed noisy self-training on a high-resource translation benchmark: WMT14 English-German and a low-resource translation benchmark: FloRes English-Nepali.

- **WMT14 English-German:** In addition to WMT100K, we also report results with all 3.9M training examples. For WMT100K we use the Base Transformer architecture, and the remaining parallel data as the monolingual data. For the full setting, we use the Big Transformer architecture (Vaswani et al., 2017) and randomly sample 20M English sentences from the News Crawl corpus for noisy ST.

- **FloRes English-Nepali:** We evaluate noisy self-training on a low-resource machine translation dataset FloRes (Guzmán et al., 2019) from English (en) to Nepali (ne), where we have 560K training pairs and a very weak supervised system that attains BLEU smaller than 5 points. For this dataset we have 3.6M Nepali monolingual instances in total (for BT) but 68M English Wikipedia sentences.[5] We randomly sample 5M English sentences for noisy ST. We use the same transformer architecture as in (Guzmán et al., 2019).

The overall results are shown in Table 4. For almost all cases in both datasets, the noisy ST outperforms the baselines by a large margin ($1 \sim 5$ BLEU scores), and we see that noisy ST still improves the baseline even when this is very weak.

**Effect of Domain Mismatch.** Test sets of the FloRes benchmark were built with mixed original-translationese – some sentences are from English sources and some are from Nepali sources. Intuitively, English monolingual data should be more in-domain with English-origin sentences and

---

[5]http://www.statmt.org/wmt19/parallel-corpus-filtering.html

| Methods | 100K (+3.7M mono) | | | 640K (+3.2M mono) | | | 3.8M (+4M mono) | | |
| --- | --- | --- | --- | --- | --- | --- | --- | --- | --- |
| | R1 | R2 | RL | R1 | R2 | RL | R1 | R2 | RL |
| MASS (Song et al., 2019)* | – | – | – | – | – | – | 38.7 | 19.7 | 36.0 |
| baseline | 30.4 | 12.4 | 27.8 | 35.8 | 17.0 | 33.2 | 37.9 | 19.0 | 35.2 |
| BT | 32.2 | 13.8 | 29.6 | **37.3** | **18.4** | **34.6** | – | – | – |
| noisy ST | **34.1** | **15.6** | **31.4** | 36.6 | 18.2 | 33.9 | **38.6** | **19.5** | **35.9** |

Table 5: Rouge scores on Gigaword datasets. For the 100K setting we use the remaining 3.7M training data as unlabeled instances for noisy ST and BT. In the 3.8M setting we use 4M unlabeled data for noisy ST. Stared entry (∗) denotes that the system uses a much larger dataset for pretraining.

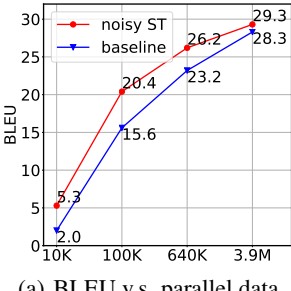
(a) BLEU v.s. parallel data

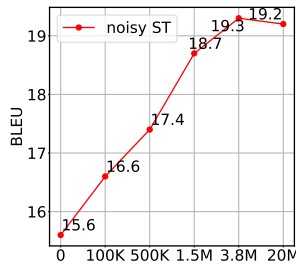
(b) BLEU v.s. monolingual data

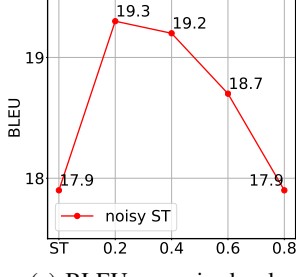
(c) BLEU v.s. noise level

Figure 3: Analysis of noisy self-training on WMT English-German dataset, demonstrating the effect of parallel data size, monolingual data size, and noise level.

Nepali monolingual data should help more for Nepali-origin sentences. To demonstrate this possible domain-mismatch effect, in Table 4 we report BLEU on the two different test sets separately.[6] As expected, ST is very effective when the source sentences originate from English.

**Comparison to Back-Translation.** Table 4 shows that noisy ST is able to beat BT on WMT100K and on the en-origin test set of FloRes. In contrast, BT is more effective on the ne-origin test set according to BLEU, which is not surprising as the ne-origin test is likely to benefit more from Nepali than English monolingual data.

## 5.3 Text Summarization

We further evaluate noisy self-training on the Gigaword summarization dataset (Rush et al., 2015) that has 3.8M training sentences. We encode the data with 30K byte-pair codes and use the Base Transformer architecture. Similar to the setting of WMT100K, for Gigaword we create two settings where we sample 100K or 640K training examples and use the remaining as unlabeled data to compare with BT. We also consider the setting where all the 3.8M parallel samples are used and we mine in-domain monolingual data by revisiting the original preprocessing procedure[7] and using the ∼4M samples that Rush et al. (2015) disregarded because they had low-quality targets. We report ROUGE scores (Lin, 2004) in Table 5. Noisy ST consistently outperforms the baseline in all settings, sometimes by a large margin (100K and 640K). It outperforms BT with 100K parallel data but underperforms with 640K parallel data. We conjecture that BT is still effective in this case because the task is still somewhat symmetric as Gigaword mostly contains short sentences and their compressed summaries. Notably, noisy ST in the full setting approaches the performance of state-of-the-art systems which use much larger datasets for pretraining (Song et al., 2019).

## 5.4 Analysis

In this section, we focus on the WMT English-German dataset to examine the effect of three factors on noisy self-training: the size of the parallel dataset, the size of the monolingual dataset, and the noise level. All the noisy ST results are after the fine-tuning step.

---

[6]Test set split is obtained through personal communication with the authors.
[7]https://github.com/facebookarchive/NAMAS

**Parallel data size.** We fix the monolingual data size as 20M from News Crawl dataset, and vary the parallel data size as shown in Figure 3(a). We use a small LSTM model for 10K, Base Transformer for 100K/640K, and Big Transformer for 3.9M.[8] Noisy ST is repeated for three iterations. We see that in all cases noisy ST is able to improve upon the baseline, while the performance gain is larger for intermediate value of the size of the parallel dataset, as expected.

**Monolingual data size.** We fix the parallel data size to 100K samples, and use the rest 3.8M English sentences from the parallel data as monolingual data. We sample from this set 100K, 500K, 1.5M, and 3.8M sentences. We also include another point that uses 20M monolingual sentences from a subset of News Crawl dataset. We report performance at the first iteration of noisy ST. Figure 3(b) illustrates that the performance keeps improving as the monolingual data size increases, albeit with diminishing returns.

**Noise level.** We have shown that noisy ST outperforms ST, but intuitively larger noise must not always be better since at some point it may destroy all the information present in the input. We adopt the WMT100K setting with 100K parallel data and 3.8M monolingual data, and set the word blanking probability in the synthetic noise (Lample et al., 2018) to 0.2 (default number), 0.4, 0.6, and 0.8. We also include the baseline ST without any synthetic noise. Figure 3(c) demonstrates that performance is quite sensitive to noise level, and that intermediate values work best. It is still unclear how to select the noise level *a priori*, besides the usual hyper-parameter search to maximize BLEU on the validation set.

### 5.5 NOISE PROCESS ON PARALLEL DATA ONLY

In this section, we justify whether the proposed noisy self-training process would help the supervised baseline alone without the help of any monolingual data. Similar to the training process on the monolingual data, we first train the model on the noisy source data (pseudo-training), and then fine-tune it on clean parallel data. Different from using monolingual data, there are two variations here in the "pseudo-training" step: we can either train with the fake target predicted by the model as on monolingual data, or train with the real target paired with noisy source. We denote them as "parallel + fake target" and "parallel + real target" respectively, and report the performance on WMT100K in Table 6. We use the same synthetic noise as used in previous experiments.

When applying the same noise process to parallel data using fake target, the smoothing effect is not significant since it is restricted into the limited parallel data space, producing marginal improvement over the baseline (+0.4 BLEU). As a comparison, 100K monolingual data produces +1.0 BLEU and the effect is enhanced when we increase the monolingual data to 3.8M, which leads to +3.7 BLEU. Interestingly, pairing the noisy source with real target results in much worse performance than the baseline (-4.3 BLEU), which implies that the use of *fake* target predicted by the model (i.e. distillation) instead of real target is important for the success of noisy self-training, at least in the case where parallel data size is small. Intuitively, the distilled fake target is simpler and relatively easy for the model to fit, but the real target paired with noisy source makes learning even harder than training with real target and real source, which might lead to a bad starting point for fine-tuning. This issue would be particularly severe when the parallel data size is small, in that case the model would have difficulties to fit real target even with clean source.

## 6 RELATED WORK

Self-training belongs to a broader class of "pseudo-label" semi-supervised learning approaches. These approaches all learn from pseudo labels assigned to unlabelled data, with different methods on how to assign such labels. For instance, co-training (Blum & Mitchell, 1998) learns models on two independent feature sets of the same data, and assigns confident labels to unlabeled data from one of the models. Co-training reduces modeling bias by taking into account confidence scores from two models. In the same spirit, democratic co-training (Zhou & Goldman, 2004) or tri-training (Zhou & Li, 2005) trains multiple models with different configurations on the same data feature set, and a subset of the models act as teachers for others.

---

[8]These architectures are selected based on validation loss.

| Methods | PT | FT |
|---|---|---|
| parallel baseline | – | 15.6 |
| noisy ST, 100K mono + fake target | 10.2 | 16.6 |
| noisy ST, 3.8M mono + fake target | 16.6 | 19.3 |
| noisy ST, 100K parallel + real target | 6.7 | 11.3 |
| noisy ST, 100K parallel + fake target | 10.4 | 16.0 |

Table 6: Results on WMT100K data. All results are from one single iteration. "Parallel + real/fake target" denotes the noise process applied on parallel data but using real/fake target in the "pseudo-training" step. "Mono + fake target" is the normal noisy self-training process described in previous sections.

Another line of more recent work perturb the input or feature space of the student's inputs as data augmentation techniques. Self-training with dropout or noisy self-training can be viewed as an instantiation of this. These approaches have been very successful on classification tasks (Rasmus et al., 2015; Miyato et al., 2017; Laine & Aila, 2017; Miyato et al., 2018; Xie et al., 2019) given that a reasonable amount of predictions of unlabeled data (at least the ones with high confidence) are correct, but their effect on language generation tasks is largely unknown and poorly understood because the pseudo language targets are often very different from the ground-truth labels. Recent work on sequence generation employs auxiliary decoders (Clark et al., 2018) when processing unlabeled data, overall showing rather limited gains.

## 7 CONCLUSION

In this paper we revisit self-training for neural sequence generation, and show that it can be an effective method to improve generalization, particularly when labeled data is scarce. Through a comprehensive ablation analysis and synthetic experiments, we identify that noise injected during self-training plays a critical role for its success due to its smoothing effect. To encourage this behaviour, we explicitly perturb the input to obtain a new variant of self-training, dubbed noisy self-training. Experiments on machine translation and text summarization demonstrate the effectiveness of this approach in both low and high resource settings.

## ACKNOWLEDGEMENTS

We want to thank Peng-Jen Chen for helping set up the FloRes experiments, and Michael Auli, Kyunghyun Cho, and Graham Neubig for insightful discussion about this project.

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

## A    EXPERIMENTS DETAILS

### A.1    SETUP DETAILS

For all experiments, we optimize with Adam (Kingma & Ba, 2014) using $\beta_1 = 0.9, \beta_2 = 0.98, \epsilon = 1e - 8$. All implementations are based on fairseq (Ott et al., 2019), and we basically use the same learning rate schedule and label smoothing as in fairseq examples to train the transformers.[9] Except for the toy sum dataset which we runs on a single GPU and each batch contains 32 examples, all other experiments are run on 8 GPUs with an effective batch size of 33K tokens. All experiments are validated with loss on the validation set. For self-training or noisy self-training, the pseudo-training takes 300K synchronous updates while the fine-tuning step takes 100K steps.

We use the downloading and preprocessing scripts in fairseq to obtain the WMT 2014 English-German dataset,[10] which hold out a small fraction of the original training data as the validation set.

The model architecture for the toy sum dataset is a single-layer LSTM with word embedding size 32, hidden state size 32, and dropout rate 0.3. The model architecture of WMT10K baseline in Figure 3(a) is a single layer LSTM with word embeddings size 256, hidden state size 256, and dropout rate 0.3.

### A.2    JUSTIFICATION OF THE WMT100K BASELINE

We provide more details and evidence to show that our baseline model on WMT100K dataset is trained properly. In all the experiments on WMT100K dataset including baseline and self-training ones, we use Adam optimizer with learning rate 0.0005, which is defaulted in fairseq. We do not use early stop during training but select the best model in terms of the validation loss. We train with 30K update steps for the baseline model and (300K pseudo-training + 100K fine-tuning) update steps for self-training. In both cases we verified that the models are trained sufficiently to fully converge through observing the increase of validation loss. Figure 4 shows the validation curve of the baseline model. Note that the model starts to overfit, and we select the model checkpoint at the lowest point. We also varied the learning rate hyperparameter as 0.0002, 0.0005, and 0.001, which produced BLEU score 15.0, 15.6 (reported in the paper), and 15.5 respectively – our baseline model in previous sections obtained the best performance.

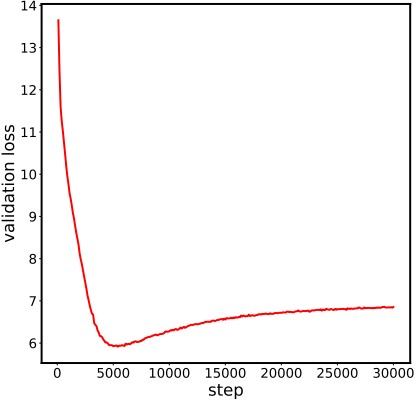

Figure 4: Validation loss v.s. number of update steps, for the baseline model on WMT100K dataset.

---

[9]https://github.com/pytorch/fairseq/blob/master/examples/translation.
[10]https://github.com/pytorch/fairseq/tree/master/examples/translation.

## B    COMPARISON REGARDING SEPARATE TRAINING, JOINT TRAINING, AND FILTERING

In the paper we perform self-training with separate pseudo-training and fine-tuning steps and always use all monolingual data. However, there are other variants such as joint training or iteratively adding confident examples. Here we compare these variants on WMT100K dataset, noisy self-training uses paraphrase as the perturbation function. For joint training, we tune the upsampling ratio of parallel data just as in back-translation (Edunov et al., 2018). We perform noisy self-training for 3 iterations, and for *filtering* experiments we iteratively use the most confident 2.5M, 3M, and 3.8M monolingual data respectively in these 3 iterations. Table 7 shows that the filtering process helps joint training but still underperforms separate-training methods by over 1.5 BLEU points. Within separate training filtering produces comparable results to using all data. Since separate training with all data is the simplest method and produces the best performance, we stick to this version in the paper.

| Methods | BLEU |
|---|---|
| baseline | 15.6 |
| noisy ST (separate training, all data) | 21.8 |
| noisy ST (separate training, filtering) | 21.6 |
| noisy ST (joint training, all data) | 18.8 |
| noisy ST (joint training, filtering) | 20.0 |

Table 7: Ablation analysis on WMT100K dataset.

## C    ADDITIONAL RESULTS ON THE TOY SUM DATASET

We additionally show the error heat maps of the entire data space on the toy sum datasets for the first two iterations. Here the model at pseudo-training step is initialized as the model from last iteration to clearly examine how the decodings change due to injected noise. As shown in Figure 5, for each iteration the pseudo-training step smooths the space and fine-tuning step benefits from it and greatly reduces the errors

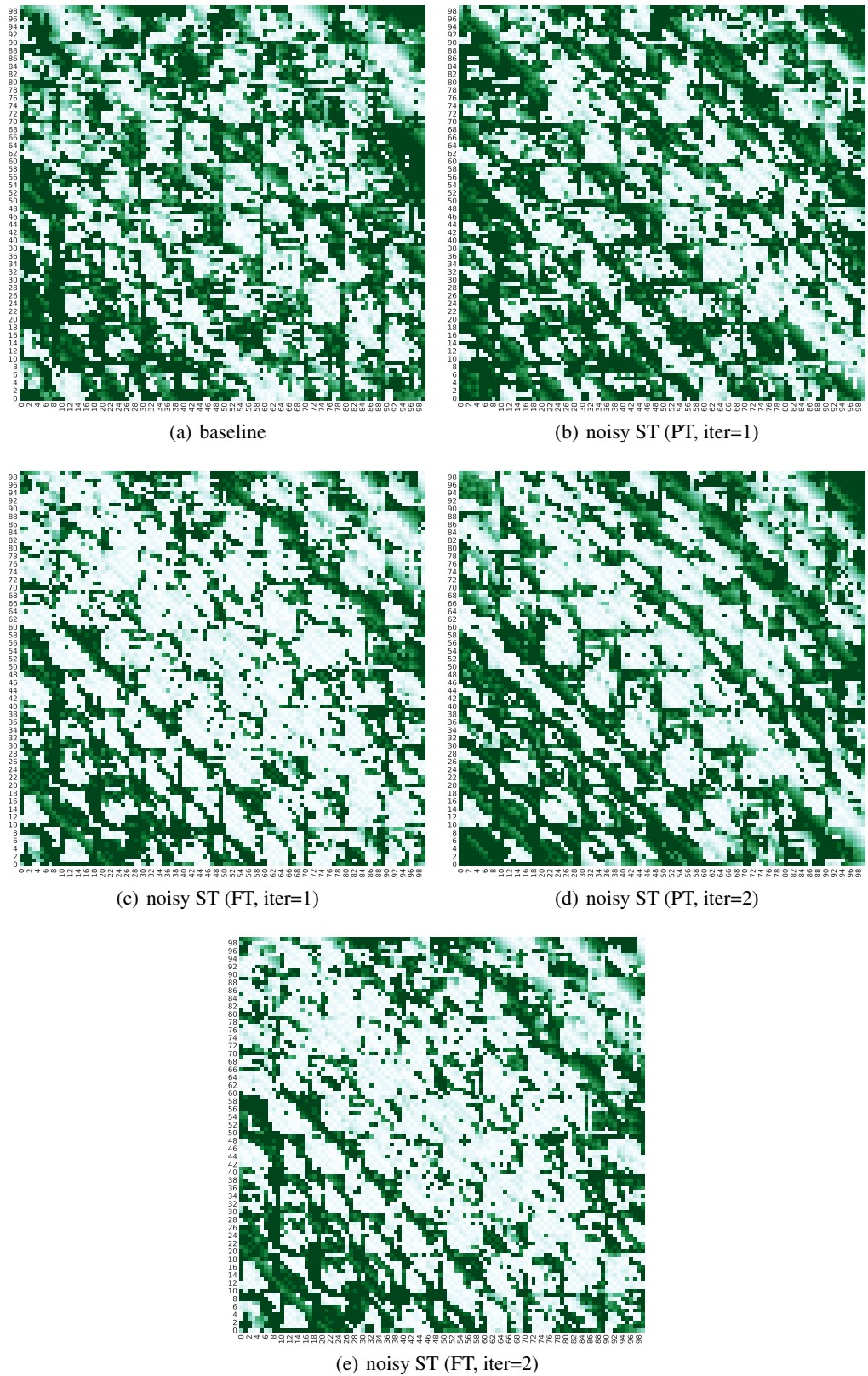

Figure 5: Error heat maps on the toy sum dataset over the first two iterations. Deeper color represent larger errors.

