# OpenReview forum: "Revisiting Self-Training for Neural Sequence Generation"
_ICLR.cc/2020/Conference — Accept (Poster)_

### Official Review · AnonReviewer3 · 2019-10-15
**Official Blind Review #3**

**Rating:** 8

**Review:**

The paper introduces an interesting study that tries to explain why conditional text generation models with autoregressive decoders benefit from self-training on pseudo labels created from the same model. The paper introduces and verifies two hypotheses: 1) Decoding strategy: Since beam search is a biased estimator sampling using it doesn't reflect the learned distribution from the model and hence variations happen that benefit learning. (this partially help).
Authors verify that by replacing the beam search decoder by an unbiased sampling method for decoding. 2) Additional noise during training: The use of dropout adds discrepancy between training and inference which creates favourable regularization. Through smoothing of the latent spaces so similar semantic inputs get mapped to similar outputs.  Authors verify that using a toy task of summing two numbers.

Following this intuition, authors introduce a new method of noisy self-training that even enforces hypothesis #2 (rather than increasing dropout which isn't practical to train). Authors experiment with word perturbation from (Lample et al. 18) and paraphrasing through translation back and forth.

Authors provide a thorough experimental section evaluating the noisy ST on Machine translation, low resource MT and Summarization the latter two latter are especially interesting to use ST rather than Back translation as target side documents might be hard to find (low resource MT) or quite challenging to recover the source side from (in case of summarization ).

Additional analysis experiments were performed to show the effect noisy self-training with regard to, increasing noise, number of available parallel data, size of ST samples generated from monolingual data.

I am in favour of this work acceptance, overall the paper introduces very interesting insights to explain the usefulness of self-training for auto-regressive generation tasks and could inspire future work along this line in designing better ST algorithms and/or adoption of self-training in low-resourced generation tasks.


Questions to authors:
As explained in section 2, you preferred to model the ST task as an iterative process between pseudo training and fine-tuning was mainly chose for simplicity while providing equal results as shown in Table 6 (appendix). I wonder why joint training does provide lower results than separate training? I doubt this might be due to that it is given less time to converge compared to the 3 iterations of self-training. Can you provide more details about how this is done?

Figure 4 (appendix), I was wondering if there's an intuition about the artefacts in the error heat maps of the toy task ( the patterns with -45 degrees slope)


Suggestions to enhance readability:
A lot of abbreviations finish by "T" this makes it quite hard to follow, I would suggest authors to remove "PT" and "FT" since they haven't been used much in the paper.

Labels in Figure 1: especially early in the paper becomes hard to grasp, would be nice if you can add description to what each abbreviation means in the figure caption.

typo table 6 appendix: joint "traing"









**Experience Assessment:**

I have published in this field for several years.

**Review Assessment: Checking Correctness Of Derivations And Theory:**

N/A

**Review Assessment: Checking Correctness Of Experiments:**

I carefully checked the experiments.

**Review Assessment: Thoroughness In Paper Reading:**

I read the paper thoroughly.

---

> ### Author Response · Authors · 2019-11-15
> **Response to Reviewer #3**
>
> Thanks for your encouraging comments and advice! Due to time limitations we could only address major points, but we’ll make sure to reflect all advice in future revisions.
>
> ## Q1:Why joint training does provide lower results than separate training ?
> We don’t fully understand the reason, but we can provide details on the training and our hypothesis. Joint training was applied for three iterations, and the performance was almost unchanged at the fourth iteration just as the separate training. However, there was an important hyperparameter that would greatly affect the performance of joint training -- the upsampling ratio, which means we need to upsample the real parallel data and mix it with the pseudo parallel data.This parameter needs to be tuned at every iteration, and we varied it as {1, 3, 5, 7, 9, 10, 20, 30} in our experiments due to resource limitations. It is possible that the performance of joint training can be improved through more search of this upsampling ratio at every iteration.
>
> ## Q2: Artefacts in the error heat maps of the toy task ( the patterns with -45 degrees slope)
> The line of -45 degree basically represents x+y=C. Since the toy sum task is actually predicting a sequence of digits, the goal becomes more difficult when every digit of C would change with subtle change of the input. For example, large errors tend to happen in the region of x+y=50 because two digits would change from 49 to 50, the line x+y=49 and x+y=50 are actually pretty close but have completely different output (in terms of edit distance), which are relatively hard to be separated by the network. In contrast, x+y=45 and x+y=44 or 46 only have one digit difference.
>
> This pattern is especially evident in Figure 5(c) after the first iteration of noisy self-training, where the error regions are around x+y=10, 20, 30, 40, 50, 60, 70, 80 ….

---

### Official Review · AnonReviewer1 · 2019-10-23
**Official Blind Review #1**

**Rating:** 6

**Review:**

This paper investigates why self-training helps in machine translation and text summarization tasks, identifying that auxiliary noise can amplify the benefits of this process. The paper is well-structured and clearly written, and conducts a fairly thorough analysis of the issue at hand.

Comments:

- The authors argue self-training enhances smoothness, but I would like to see this explained mathematically/conceptually in greater detail. It is not immediately clear to me why this would be, particularly in the case of discrete text data.

- Why not evaluate smoothness on the actual MT task instead of just the toy task?
The authors could measure the L2 norm between encodings of source sentences for neighboring sentences (eg. based on edit distance or word-movers) vs very different sentences. And then compare the base model vs the one obtained from self-supervised training.

- If the primary beneficial effect of self-supervised training is smoothness as the authors claim, then they should try enforcing smoothness in alternative ways to see if performance improves.  Some options here could be (using dropout in all of them): 1) add your same noise process to the original labeled dataset to create augmented examples, 2) rather than the self-supervised objective, use an auxiliary training objective which says the predictions on each unlabeled datapoint should be similar to the predictions on noised versions of this datapoint, 3) some form of virtual adversarial training [VAT]. In fact, the authors should discuss [VAT] a bit more, as this paper also presents a smoothness-regularizer that is highly useful for semi-supervised learning.

[VAT] Miyato et al. Virtual Adversarial Training: A Regularization Method for Supervised and Semi-Supervised Learning. https://arxiv.org/abs/1704.03976


- As the authors write, self-training can be viewed as a form of entropy-regularization. Likewise, the input perturbation+dropout process also seems like it should affect the conditional entropies. Can the authors expound upon this connection a bit further?  Some mathematical analysis would be nice to have here as well.


- "Another way is to treat them separately – first we train the model on pseudo parallel data S, and then fine-tune it on real data L."

The authors should clarify the overall training process with a more precise explanation. I assume is actually:
1) train initial model M on (limited) real data L
2) use M to generate pseudo-targets for unlabeled data in S
3) train M on this pseudo-dataset S
4)fine-tune M on real data L

Is this correct? And it should be clarified whether M in step (3) is fine-tuned from the M in step (1) or re-initialized from scratch before training begins (Based on later text, it seems like the latter, but this should be clarified early on).

- Baseline in Figure 1 should be described a bit more clearly.

- Since the BLEU score dropped from 3 to 1.9 when the authors continued training from the baseline model (Sec 3.2), isn't the optimum hypothesis not ruled out?  I don't think the authors should make this claim, and rather state that the initialization does seem to play some role, but does not fully explain the benefits of self-training.

- "We use a small LSTM model for 10K, Base Transformer for 100K/640K, and Big Transformer for 3.9M"

Is this because these are the best performing models on these respective datasets? The authors should explain these decisions.


- "We include quantitative comparison regarding joint training, separate training, and pseudo-parallel data filtering in Appendix B"

Should clarify here (in main text) that separate training matches the performance of joint training.

- typos: "joint traing"

**Experience Assessment:**

I have published one or two papers in this area.

**Review Assessment: Checking Correctness Of Derivations And Theory:**

N/A

**Review Assessment: Checking Correctness Of Experiments:**

I carefully checked the experiments.

**Review Assessment: Thoroughness In Paper Reading:**

N/A

---

> ### Author Response · Authors · 2019-11-15
> **Response to Reviewer #1**
>
> We thank the reviewer for the time and comments. Due to time limitations we could only address major points, but we’ll make sure to reflect all advice in future revisions.
>
> ## Q1: Noise process on the original labeled dataset
> This is a good point. We have updated paper and added this experiment as Section 5.5 (please see the paper for details). In short, using the same noisy training process on the original labeled data without any monolingual data produces marginal improvement over the baseline (+0.4 BLEU) on WMT100K. The smoothing effect in this case is not significant since it is restricted into the limited parallel data, compared with more monolingual data (3.8M instances) that produces 3.7 BLEU improvement.
>
>
> ## Q2: Auxiliary training objective which says the predictions on unlabeled and noisy data should be similar ?
> Thank you for bringing this up! This is a good point, and we think the noisy self-training objective is already a form of such objectives. Specifically, the training objective of pseudo-training stage of self-training can be understood as minimizing the following KL distance:
>
> $$KL(p_{\theta}(\cdot|x) || p_{\theta^{\prime}}(\cdot|x’)) = \mathbb{E}_{y\sim p_{\theta}(\cdot|x)}[\log p_{\theta}(y|x) - \log p_{\theta^{\prime}}(y|x’)],$$
>
> where x is the clean monolingual data and x’ is the noisy one, this KL distance is between the output distributions of unlabeled points and their noisy versions. If we do not propagate the gradients through $\theta$ (as also suggested by the [VAT] paper) but only update $\theta^{\prime}$, then pseudo-training in self-training is equivalent to minimizing this KL distance, which is to force the predictions from two distributions to be similar to each other, as the reviewer suggested in this question.
>
> ## Q3: Self-training as form of entropy-regularization
> Thank you for the advice! It is true that the input perturbation+dropout process affects the conditional entropy equation. As an alternative of expanding this with more mathematical analysis, we are currently thinking of explaining noisy self-training as minimizing the KL distance as shown in the response to Q2. We will keep thinking and add the mathematical connections in the next revision. Also, we would really appreciate it if you have other ideas or suggestions to connect this with conditional entropy.
>
> ## Q4: Clarification of the overall training process
> Your understanding is correct and sorry for the confusion. During pseudo-training the model M is re-initialized from scratch. We have updated the paper and clarified this in Section 3.2 when we first introduced the fine-tuning step.
>
> ## Q5: Isn’t the optimum hypothesis not ruled out since the BLEU score dropped from 3 to 1.9 ?
> We want to clarify that the 3 BLEU improvement of ST is after three iterations from Figure 1, while the 1.9 BLEU improvement is from one single iteration from Table 1. As shown in Table 1, random initialization and continuing training produced similar results, thus we ruled out the optimum hypothesis.
>
> ## Q6: Some form of virtual adversarial training and more discussion of the [VAT] paper
> Thank you for pointing out the paper. Virtual adversarial training is proven super useful on classification tasks through perturbing towards the adversarial direction. However, perturbing text towards the adversarial direction is a bit tricky due to the discreteness, the [text VAT] paper instead perturbed the word embeddings for text classification. We didn’t include the form of VAT in the experiments since it is much more complicated than the noisy self-training used in the paper. We will definitely include more discussion about VAT, and consider experimenting with the adversarial perturbation in the next revision.
>
> ## Q7: Evaluating smoothness on the actual MT task
> This is a good point. We chose the toy task because the distance between the numeric data points is more promising to be trusted compared with the edit distance between sentences. However, we agree that evaluating on the actual MT task would be definitely more convincing, we will try to add this experiment in the next revision.
>
> ## Q8: Should clarify (in main text) that separate training matches the performance of joint training.
> Thank you for the advice! We have updated the paper to clarify this.
>
> ## Q9: Explanation of decisions for “we use a small LSTM model for 10K, Base Transformer for 100K/640K, and Big Transformer for 3.9M”.
> Thank you for noting this! We have added explanation in the paper. In short, it is difficult to train a transformer on small data like 10K (which overfits very quickly), thus we use LSTM and tune its hyperparameters. For 100K/640K/3.9M, we chose between base and big transformers and selected with validation loss.
>
>
> [VAT] Miyato et al. Virtual Adversarial Training: A Regularization Method for Supervised and Semi-Supervised Learning.
> [text VAT] Miyato et al. Adversarial Training Methods for Semi-Supervised Text Classification

---

### Official Review · AnonReviewer2 · 2019-10-23
**Official Blind Review #2**

**Rating:** 6

**Review:**

This paper presents a self-training approach for improving sequence-to-sequence tasks. As a preliminary experiment, this study randomly sampled 100k sentences from WMT 2014 English-German dataset (WMT100K, hereafter), trained a baseline (Transformer) model on WMT100K, and applied self-training methods on the remaining English sentences as the unlabeled monolingual data. After exploring different procedures for self-training, this study uses the fine-tuning strategy: train a model on the supervision data; build pseudo parallel data by predicting translations for all unlabeled data using the trained model; train a new model on the pseudo parallel data; and fine-tune the new model on the supervision data. This strategy alone gave a 3 points improvement of BLEU.

This paper hypothesized the reasons behind the performance improvements: beam-search decoding (+0.5 BLEU) and dropout (+1.2 BLEU). The authors argue that the beam-search decoding contributes partially to the performance gains, while the dropout accounts for the most. The authors infer that the dropout causes an implicit perturbation. Exploring different perturbation strategies, synthetic noise (e.g., input tokens are randomly dropped, masked, and shuffled) and paraphrase (round-trip translation, e.g., English-German-English), the authors reported no significant difference between these two strategies. Finally, this paper reports empirical results (on WMT 2014 English-German, FloRes English-Nepali, and English Gigaword (summarization task)) of self-training strategies presented in this paper. This paper concluded that self-training can be an effective method to improve generalization and that the noise injected during self-training plays a critical role for its success.

This paper is well structured and well written. It is interesting to see the improvements of the sequence-to-sequence tasks by using the self-training approach. This paper is interesting because it has a close connection with the back-translation approach that has been popular in recent years.

Although the hypotheses presented in this paper are interesting, they were not fully validated after all. The analyses on the loss functions, ablation tests, and experiments on the toy task can only bring indirect explanations about why we could observe the performance improvements. This impression is also demonstrated by the fact that this paper uses 'might' five times when explaining interpretations of the experimental results. Having said, I tend to agree that identifying the exact reason is difficult.

However, I have two other questions before recommending this paper: (1) whether the baseline model was trained sufficiently, and (2) whether this paper is about self-training strategies or regularization methods.

(1) In order to accept the experimental results that the self-training approach can improve the performance, we need to make sure that the baseline model was trained sufficiently. However, the appendix explains, "we basically use the same learning rate schedule and label smoothing as in fairseq examples." I'm not sure whether this training procedure was fair among different models because the baseline model and self-supervised model received totally different number of training instances (100k vs 3.9M). This claim would be stronger if this paper could show an evidence that the baseline model was trained properly by, for example, explaining the stopping criteria for iterations, tuning hyper-parameters (e.g., learning rate) individually for the baseline and self-trained models, showing the mean and variance of BLEU scores with different initializations, and showing the training curve of the baseline model.

(2) We can view the self-training strategies presented in this paper as regularization methods. For this reason, I am wondering whether the self-training strategies presented in this paper are only for self-training or general to pure supervised setting. We can easily guess that the performance would drop if we removed the dropout from the baseline method. In contrast, I would like to see whether the synthetic noise could improve the performance of the baseline method alone, behaving as a regularization method. It would be useful to see the performance of the baseline method without the dropout and with the synthetic noise to highlight the effect of the presented strategies under the self-training scenario.

Minor comments:

It would be useful to see the number of unlabeled instances in Section 3.1.

Sectoin 3.2: "This is different from back-translation where new knowledge may originate from an additional backward translation model."
I'm not sure whether a backward translation model can introduce new knowledge because the supervision data are usually the same between forward and backward directions.

Section 3.3: "at test/decoding time the model does not use dropout"
To be precise, the weights are scaled by the dropout rate (p) in the decoding time.

Reference: Lample et al. 2018 should be replaced with another paper:
Guillaume Lample, Alexis Conneau, Ludovic Denoyer, Marc'Aurelio Ranzato. Unsupervised Machine Translation Using Monolingual Corpora Only. ICLR 2018.

**Experience Assessment:**

I have read many papers in this area.

**Review Assessment: Checking Correctness Of Derivations And Theory:**

I carefully checked the derivations and theory.

**Review Assessment: Checking Correctness Of Experiments:**

I carefully checked the experiments.

**Review Assessment: Thoroughness In Paper Reading:**

I read the paper thoroughly.

---

> ### Author Response · Authors · 2019-11-15
> **Response to Reviewer #2**
>
> We thank the reviewer for the time and comments. Due to time limitations we could only address major points, but we’ll make sure to reflect all advice in future revisions. We also appreciate the reviewer’s understanding that fully validating the actual reason is difficult, we will keep working on this and hopefully to provide more direct experiments or analysis to have a deeper understanding of the principle in the next revision.
>
>
> ## Q1:Whether the baseline model was trained sufficiently
> Thank you for bringing this up! First, we want to clarify that the baseline models for WMT with full 3.9M training data, FloRes, and Gigaword summarization with full 3.8M training data are trained properly because our baselines are able to match performance from previous work on these benchmarks.
>
> We believe the main concern of the baseline training is on WMT100K. We updated the paper and justified this in Appendix A.2 (please see paper for details). The baseline was trained with 30K updates without early stopping and we show the curve of validation loss which implies that the model already converges. The model checkpoint with the best validation loss was selected. We also varied the learning rate as 0.0002, 0.0005 (this is the learning rate we used in all the experiments), and 0.001 in Appendix A.2, our baseline model with learning rate 0.0005 actually performs the best. Tuning other hyperparameters and reporting mean/variance of BLEU needs more time, we will keep running these experiments to verify this 100K baseline and add them in the next revision.
>
>
> ## Q2: Whether this paper is about self-training strategies or regularization methods.
> This is a good point. We have updated the paper and added the suggested experiments as Section 5.5 (please see paper for details). In short, the performance of baseline without dropout is very bad (BLEU score is 5.2 on WMT100K, this result is added to Table 2). Using the same noisy training process on the original labeled data without any monolingual data produces marginal improvement over the baseline (+0.4 BLEU) on WMT100K. The noise effect in this case is not significant since it is restricted into the limited parallel data, compared with much more monolingual data (3.8M instances) that produces 3.7 BLEU improvement.

---

### Author Response · Authors · 2019-11-15
**Revision Submitted**

We have submitted a revised manuscript and made the following modifications to address the reviewers' major concerns:

-- Applied the same noise process to supervised data without the help of monolingual data (Section 5.5)
-- Justification of whether the baseline model on WMT100K data is trained sufficiently or not (Appendix A.2 )


While limited by time in the response period, we do still plan to address *all* the reviewer’s comments in future revisions. We also welcome any further feedbacks to improve this paper !

---

### Decision · Program_Chairs · 2019-12-19

**Decision:**

Accept (Poster)

**Comment:**

This paper analyzes self-training for sequence-to-sequence models and proposes a noisy version of self training. An empirical study shows the proposed noisy version improves results for machine translation and summarization tasks.

All reviewers appreciate the interesting contributions of the research, as well as clear writing. They also offer several comments for the revision of the paper.

We look forward to seeing this paper presented at the conference!